# Ferroptosis-Related Molecular Clusters and Diagnostic Model in Rheumatoid Arthritis

**DOI:** 10.3390/ijms24087342

**Published:** 2023-04-16

**Authors:** Maosheng Xie, Chao Zhu, Yujin Ye

**Affiliations:** Department of Rheumatology and Immunology, The First Affiliated Hospital, Sun Yat-sen University, Guangzhou 510080, China

**Keywords:** rheumatoid arthritis, ferroptosis, molecular clusters, immune, predictive model

## Abstract

Rheumatoid arthritis (RA) is a systemic autoimmune disease characterized by synovitis, joint damage and deformity. A newly described type of cell death, ferroptosis, has an important role in the pathogenesis of RA. However, the heterogeneity of ferroptosis and its association with the immune microenvironment in RA remain unknown. Synovial tissue samples from 154 RA patients and 32 healthy controls (HCs) were obtained from the Gene Expression Omnibus database. Twelve of twenty-six ferroptosis-related genes (FRGs) were differentially expressed between RA patients and HCs. Furthermore, the patterns of correlation among the FRGs were significantly different between the RA and HC groups. RA patients were classified into two distinct ferroptosis-related clusters, of which cluster 1 had a higher abundance of activated immune cells and a corresponding lower ferroptosis score. Enrichment analysis suggested that tumor necrosis factor-α signaling via nuclear factor-κB was upregulated in cluster 1. RA patients in cluster 1 responded better to anti-tumor necrosis factor (anti-TNF) therapy, which was verified by the GSE 198520 dataset. A diagnostic model to identify RA subtypes and immunity was constructed and verified, in which the area under the curve values in the training (70%) and validation (30%) cohorts were 0.849 and 0.810, respectively. This study demonstrated that there were two ferroptosis clusters in RA synovium that exhibited distinct immune profiles and ferroptosis sensitivity. Additionally, a gene scoring system was constructed to classify individual RA patients.

## 1. Introduction

Rheumatoid arthritis (RA) is a common autoimmune disease characterized by synovitis, with extra-articular manifestations such as pericarditis and interstitial lung disease in some patients [1]. Several risk factors, including genetics, smoking and the microbiota, contribute to the onset and development of RA [2]. Activation of Toll-like receptors (TLRs) by microbial and endogenous ligands has been implicated in the pathogenesis of RA [3]. Studies have also extensively examined the role of infections such as Epstein–Barr virus and Mycobacterium avium subsp. paratuberculosis in the dysregulation of the interferon regulatory factor 5 (IRF5) pathway and their association with RA [4,5]. In addition, macrophages, which play a crucial role in the progression of RA [6], exhibit features of ferroptosis in Mycobacterium tuberculosis-infected macrophages [7]. Regional differences in RA have led to an incidence that ranges from 0.5% to 1% [8]. Despite the widespread use of conventional and biological disease-modifying anti-rheumatic drugs, between 5% and 20% of refractory RA patients show no response to any of the current treatments [9]. Improvement in therapeutic efficacy has been a primary objective in the management of RA. A crucial factor contributing to the unsatisfactory treatment efficacy is the heterogeneity of pathogenic mechanisms among RA patients. Therefore, exploratory research to identify new mechanisms and subtypes at the molecular level may lead to the individualized treatment of RA patients.

Ferroptosis is a type of cell death that was first proposed in 2012 and differs from apoptosis, pyroptosis and programmed necrosis [10]. Key features of ferroptosis include the accumulation of iron and lipid peroxidation. Elevated iron levels are related to oxidative damage and ultimately to ferroptosis [11]. Lipid peroxidation is the main cause of ferroptosis, and its products can destroy the phospholipid bilayer and induce cell membrane rupture [12]. Additionally, it involves remarkable morphological changes in mitochondria, which include reduction of mitochondrial volume and cristae, increased membrane density and disruption to the mitochondrial membrane structure [13,14]. Ferroptosis is involved in non-neoplastic and neoplastic diseases [15,16,17]. A recent study suggested that ferroptosis was decreased in the synovium and synovial fibroblasts (SFs) in RA patients, and glycine alleviated disease progression in collagen-induced arthritis (CIA) mice by increasing ferroptosis [18]. However, another study revealed that there was increased lipid peroxidation and iron in the synovium and synovial fluid of RA patients. Furthermore, patients with higher disease activity had more lipid peroxidation and iron levels in inflamed joints than those with lower disease activity [19]. Galectin-1-derived peptide 3 was shown to induce ferroptosis in SFs through the p53/ solute carrier family 7 member 11 (SLC7A11) axis [20]. Several in vivo studies also demonstrated that a ferroptosis inducer ameliorated arthritis in a CIA mouse model [18,19]. Meanwhile, inhibiting ferroptosis in chondrocytes may also improve the arthritis scores and bone erosion in adjuvant arthritis rats [21]. However, the role of ferroptosis in RA remains controversial and requires further exploration.

In this study, we comprehensively evaluated the expression of ferroptosis-related genes (FRGs) in RA patients and healthy controls (HCs) and analyzed differences between the two groups. Based on differential expression of 26 FRGs, we classified 154 RA patients into two ferroptosis-related subtypes, each of which exhibited different immune states. Finally, to improve the precision of treatment, we established a scoring system to quantify the ferroptosis-related pattern in individual patients.

## 2. Results

### 2.1. FRGs Differentially Expressed between RA Patients and HCs

The study workflow is shown in Figure 1. First, we conducted an overall evaluation of the included data. All details of the data pre-processing are shown in Appendix A. We performed principal component analysis (PCA) to evaluate the effect of batch removal, and no significant outlier samples were found in a cluster analysis. There was no significant difference in the overall distribution trend in a boxplot analysis. Taken together, the data pre-processing results revealed that our data were eligible for subsequent analyses.

To determine whether ferroptosis is involved in the pathogenesis of RA, we evaluated the expression of 12 FRGs that were differentially expressed between the RA and HC groups and found that six (*ATG5*, *FTH1*, *GSS*, *NCOA4*, *TFRC* and *TP53*) were upregulated, and six (*ACSL1*, *GPX4*, *LPCAT3*, *MAP1LC3B*, *SLC3A2* and *TF*) were downregulated in the RA group (Figure 2A). Furthermore, analysis of correlations among the 26 FRGs revealed that RA patients had a diverse correlation pattern compared with that of HCs (Figure 2B–D).

### 2.2. Identification of Ferroptosis Phenotypes in RA Patients

Because FRGs are known to play important roles in RA, to further explore the subtypes of ferroptosis in RA, we classified 154 RA patients using the unsupervised cluster method. The consensus matrix showed that RA patients could be classified into two distinct clusters that exhibited different FRG expression profiles (Figure 3A,B). Additionally, the relative change in the area under the cumulative distribution function (CDF) curve was significantly different when k = 2, which was determined to be the most appropriate number (range, 2–10) based on the Consensus Cluster approach. Consequently, we divided RA patients into two clusters: 102 cases in cluster 1 and 52 cases in cluster 2. PCA showed a clear separation between these two clusters (Figure 3C–E). 

### 2.3. FRG Expression and Ferroptosis Scores in the Two Ferroptosis Clusters

After identifying two ferroptosis-related subtypes, we first compared FRG expression and ferroptosis scores in the two cluster groups. As shown in Figure 4A, cluster 1 had higher expression of *ACSL4*, *ACSL6*, *ALOX15*, *FTH1*, *GSS*, *HMOX1*, *SAT1*, *SLC7A11* and *TFRC*, while cluster 2 had higher expression of *ACSL3*, *ATG5*, *GCLC*, *GPX4*, *PCBP1*, *SLC39A14*, *SLC3A2*, *TF*, *VDAC2* and *MAP1LC3B* (*p* < 0.05). These results suggested that the two clusters had different ferroptosis patterns. Furthermore, a PCA method revealed that Cluster 1 exhibited a relatively lower ferroptosis score (Figure 4B).

### 2.4. Immune Landscape Characteristics in the Two Ferroptosis Clusters

Given that RA is a type of inflammatory arthritis, we aimed to investigate possible differences in the immune environment between the two ferroptosis clusters. Comparison of the relative proportions of 22 immune cell types using the CIBERSORT method revealed that cluster 1 had relative higher infiltration of plasma cells, activated CD4+ memory T cells, gamma delta T cells, monocytes, M1 macrophages and neutrophils. Meanwhile, cluster 2 had relative higher infiltration of resting memory CD4+ T cells, activated natural killer (NK) cells and resting mast cells (Figure 5A). The ESTIMATE algorithm was used to further quantify the level of immune infiltration in ferroptosis clusters. Cluster 1 had a significantly higher immune score, which was consistent with the higher proportion of activated immune cells (Figure 5B,C). 

### 2.5. Biological Enrichment Analysis to Further Distinguish the Ferroptosis Clusters

To elucidate biological differences between the two ferroptosis clusters, we performed Gene Set Variation Analysis (GSVA) analysis and found that tumor necrosis factor-α (TNF-α) signaling via nuclear factor-κB (NF-κB) and apoptosis were upregulated in cluster 1, while TGF-β signaling and cholesterol homeostasis were upregulated in cluster 2 (Figure 6A). Additionally, Kyoto Encylopedia of Genes and Genomes (KEGG) analysis showed that p53 signaling, glutathione (GSH) metabolism and apoptosis were also upregulated in cluster 1 (Figure 6B).

### 2.6. Relationship between the Ferroptosis Clusters and Clinical Response

External validation was conducted to confirm the clinical relevance of ferroptosis clustering. The results revealed that among the 33 patients with RA, 17 were classified into cluster 1, while 16 were classified into cluster 2, based on 26 FRGs (Figure 7A). Moreover, among the 19 good responders, 16 patients belonged to cluster 1, while 13 of the 14 non-responders were in cluster 2 (*p* = 5.68 × 10^−5^), indicating that patients in cluster 1 may respond better to anti-TNF therapy (Figure 7B).

### 2.7. Ferroptosis Cluster-Related DEGs and Construction of a Diagnostic Model

Our previous work suggested that there were different ferroptosis patterns and types of immune status in RA patients. To further distinguish the RA subtypes and thereby improve treatment, we constructed a diagnostic model. Among the 209 DEGs between the two ferroptosis clusters, 81 were upregulated in cluster 1 and 128 were upregulated in cluster 2. The clusters exhibited significantly different gene expression profiles (Figure 8). First, a protein–protein interaction (PPI) network based on the 209 DEGs was constructed using STRING. Next, 13 hub genes with a degree above 20 identified through the Cytoscape plugin CytoHubba (Figure 9A) were considered as candidate genes for least absolute shrinkage and selection operator (LASSO) regression analysis. In the cross-validation process, lambda-min was regarded as the optimal value, from which we obtained calculated regression coefficients (Figure 9B). Finally, nine hub genes were selected and used to construct a diagnostic model. Differences in the degrees of expression between the two clusters are displayed as a heatmap in Figure 8. The risk score was calculated as follows: Risk score = (−0.0747 × *MMP9*) + (−0.0871 × *CCL5*) + (−0.1251 × *ITGAX*) + (−0.0722 × *SERPINE1*) + (−0.1517 × *FCGR3B*) + (−0.0706 × *CXCL10*) + (−0.0373 × *MMP1*) + (0.1967 × *ITGB2*) + (−0.0109 × *COL1A1*). Further analysis showed that cluster 2 had higher risk scores in both the training (*n* = 102) and validation (*n* = 52) cohorts (Figure 9C,D). Additionally, the area under the curve (AUC) values of the training and validation cohorts were 0.849 and 0.810, respectively, indicating an excellent performance of this diagnostic model (Figure 9E,F), which contributes to the categorization of RA patients by degree of ferroptosis. Finally, we show heatmaps of the expression of the nine hub genes in the two ferroptosis clusters (Figure 9G,H). Furthermore, Gene Ontology (GO) and KEGG enrichment analyses based on these nine hub genes revealed that the DEGs were enriched in complement and coagulation cascades, rheumatoid arthritis, interleukin (IL)-17 and TNF signaling pathways (Figure 10).

## 3. Discussion

Currently, the treatment of RA does not depend on pathology, in contrast to the approach used to classify and treat tumors. Some researchers have proposed new RA classification systems based on disease heterogeneity. A previous study identified three pathotypes in RA synovium (lympho-myeloid, diffuse-myeloid and pauci-immune) that closely related to clinical characteristics [22]. RA patients were also classified into three N6-methyladenosine (m6A) modification clusters, in which lower m6A scores were associated with better response to infliximab therapy [23]. Previous studies demonstrated that cell subtypes and pathways were significantly different between blood and synovium from RA patients, and that it was inappropriate to use RA blood as a substitute for the tissue sites of inflammation [24]. In this study, using published synovium transcriptome sequencing data from several common datasets, we found differential expression of FRGs between RA patients and HCs. Classification of the RA patients into two subtypes according to FRG expression and further analysis indicated that cluster 1 had higher immune scores coupled with lower ferroptosis scores. Our external validation results suggested that RA patients in cluster 1 had a better response to anti-TNF therapy. Finally, we constructed a diagnostic model to evaluate individual patients.

As typical systemic autoimmune diseases, neutropenia is associated with neutrophil ferroptosis in patients with systemic lupus erythematosus, whereas peripheral neutrophil numbers are higher in patients with RA than in HCs, which implies a different pathogenesis [25]. In RA patients, we have paid more attention to the relationship between ferroptosis and synovium, especially SFs [26]. In addition, RA patients are known to have an increased risk of cardiovascular disease, and recent evidence suggests that ferroptosis may contribute to this heightened risk [27,28]. The mechanism still needs to be further explored. In this study, twelve FRGs were differentially expressed between the synovium of RA patients and HCs. These genes participated in key mechanisms of ferroptosis, including GSH metabolism, iron metabolism and lipid metabolism [29]. Meanwhile, there were apparent differences in the correlations between FRGs in RA patients and HCs. These results suggested that ferroptosis may play an important role in the onset and development of RA. Considering the heterogeneity of RA, we further classified RA patients into two distinct ferroptosis subtypes that were validated by PCA. We found that most FRGs were differentially expressed between the ferroptosis clusters and that patients in cluster 1 had lower ferroptosis scores. Considering that immune disorder is the key feature of RA, there are abnormal levels of immune cells in blood and synovium in patients [30]. Therefore, we compared the infiltration of immune cells in the two clusters. Cluster 1 had higher infiltration of plasma cells, activated CD4+ memory T cells, gamma delta T cells, monocytes, M1 macrophages and neutrophils. Plasma cells produce characteristic autoantibodies including rheumatoid factor and anti-cyclic citrullinated peptide [31]. Activated CD4+ memory T cells produce many cytokines, ultimately leading to joint damage in RA [32]. Monocytes and macrophages are important components of innate immunity. Circulating CD14+ monocytes differentiate into macrophages in RA synovium, and maintain high inflammatory and metabolic features [33]. Neutrophils in synovial fluid have been shown to secrete inflammatory mediators, and also to promote the proliferation of B cells and the production of autoantibodies [34]. These results are consistent with our observation of higher immune scores in cluster 1 patients.

The negative relationship between immune activation and lower ferroptosis score was also observed in lung adenocarcinoma [35]. In our study, cluster 1 patients had higher proportional infiltration of macrophages and activated T cells, which can produce TNF [2]. Hallmark pathway enrichment confirmed that TNF-α signaling via NF-κB was upregulated in cluster 1. As a central cytokine, TNF plays an important role in the pathogenesis of RA, and anti-TNF treatment greatly improves the prognosis of patients [36,37]. Regarding the relationship between TNF and ferroptosis in RA synovium, a recent study revealed that TNF promoted cellular GSH and protected SFs from ferroptosis through NF-κB signaling [19]. TNF can induce the upregulation of SLC7A11 in SFs; as shown in Figure 4, *SLC7A11* is upregulated in cluster 1 [19]. It is responsible for the generation of intracellular GSH, and GSH can reduce lipid peroxides to maintain cellular homeostasis [38]. SFs are major cellular elements in RA synovium, which may explain the lower ferroptosis scores in cluster 1. In the external validation, our results suggested that RA patients in cluster 1 responded better to anti-TNF therapy, which confirmed that there were some meaningful differences between the two clusters. 

To better distinguish the two RA subtypes, we constructed a diagnostic model that could be quantified in individual patients. It performed well in both the training and validation cohorts. A previous study established a risk model based on prognostic DEGs between cancer stem cell subtypes in gastric cancer [39]. In this model, patients with lower risk scores were classified into cluster 1, which had the combined feature of higher immune activation and lower ferroptosis score compared with cluster 2. In another study, RA patients with immune activation at baseline had a better response to anti-TNF treatment [40]. Because a TNF antagonist has been shown to sensitize SFs to ferroptosis, a combination of anti-TNF therapy and ferroptosis inducer might be expected to improve synovitis and cartilage damage in collagen-induced arthritis [19]. Considering that the two ferroptosis clusters have significantly different characteristics, our model may bring some help to the individualized treatment of RA patients.

There are still some limitations in this study. Firstly, our findings were based on bioinformatics analysis. Although we conducted external validation, the connection between ferroptosis clusters and clinical aspects remained unclear in RA patients. Secondly, we combined and analyzed multiple samples, and despite efforts to eliminate batch effects, there might still be confounding factors that could affect the results. Finally, the efficacy of the model needs to be further evaluated in patients with RA.

## 4. Materials and Methods

### 4.1. Data Collection and Preprocessing

Eight Gene Expression Omnibus (GEO) datasets (GSE 153015, GSE 1919, GSE 36700, GSE 55235, GSE 55457, GSE 55584, GSE 77298 and GSE 48780) related to RA were retrieved from the GEO database (https://www.ncbi.nlm.nih.gov/geo/, accessed on 2 October 2022). They included synovial tissue of 154 RA patients and 32 HCs in total. Detailed information is shown in Table 1. In particular, GSE153015 included 20 paired synovial samples of small and large joints from 10 RA patients, and cellular and molecular alterations in RA synovitis were similar between small and large joints from the same patient [41]. For the purpose of one-to-one correspondence between each sample and the patient, and decrease of the heterogeneity with other studies, we removed the duplicated samples from the same patient and finally, 10 synovial samples from large joints were chosen. Next, we processed original data from each platform through the affy R package, including background correction and standardization. Then, we used the sva R package to correct the batch effect of datasets. When multiple probes corresponded to the same gene, the maximum expression was taken. Ultimately, 8931 genes were identified.

### 4.2. Expression and Interactions of Ferroptosis-Related Genes

First, 41 genes were obtained from the ferroptosis pathway map of the KEGG database. By further screening in the FerrDb database (http://www.zhounan.org/ferrdb), PubMed and Google Scholar databases, 30 genes were identified that acted as ferroptosis drivers or suppressors [42]. After eliminating four of these (*ATG7*, *MAP1LC3A*, *SLC40A1* and *FTMT*) because they were not detected in our initial screening, 26 FRGs (16 drivers and 10 suppressors) were used for further analyses. We evaluated differences in FRG expression between RA patients and HCs, and also conducted a correlation analysis of the FRGs.

### 4.3. Identification of Ferroptosis Clusters in RA

The ConsensusClusterPlus R package was used to classify RA patients into different clusters using an unsupervised clustering method. The optimal number of categories was determined by sum of squared errors (SSE). RA patients were classified using K-means clustering combined with PCA.

### 4.4. Differentially Expressed Genes (DEGs) between Ferroptosis Clusters

DEGs between ferroptosis clusters were screened with the limma R package using cut-off values of |log1.5 fold change (FC)| > 1 and *p* < 0.05. We also used the ggplot2 and pheatmap R packages to draw volcano maps and heatmaps for visualization of DEGs.

### 4.5. Calculation of the Ferroptosis Score

After analyzing the differential expression of FRGs between ferroptosis clusters, we used these DEGs to perform PCA on RA samples to extract principal component (PC) scores of PC1 and PC2. Considering the heterogeneity of ferroptosis modification patterns in RA patients, we defined an indicator to establish a scoring system to comprehensively quantify this, which was termed the ferroptosis score. A higher ferroptosis score indicated a relative higher sensitivity to ferroptosis. Finally, a method similar to the gene expression grade index (GGI) was applied to construct a ferroptosis score [43] where ‘*i*’ indicates the expression of FRGs.
$$\mathrm{Ferroptosis}\;\mathrm{Score}=\sum \left({PC1}_{i}+{PC2}_{i}\right)$$

### 4.6. Immune Characteristics in Each Ferroptosis Cluster

Using the LM22.txt file, which contains the characteristic immune gene expression profiles of 22 immune cell types, we applied the CIBERSORT algorithm to evaluate differences in the abundance rates of these 22 types between the ferroptosis clusters. The estimate R package was used to calculate immune scores for each RA patient. Finally, we evaluated differences in the expression of several typical innate immune system genes between the ferroptosis clusters.

### 4.7. Functional and Pathway Enrichment Analysis

The gene sets “h.all.v7.2.symbols” and “c5.go.bp.v7.2.symbols” were downloaded from the MSigDB database for GSVA enrichment analysis; pathways showing a *p* value < 0.05 were defined as significantly enriched. The clusterProfiler R package was used to assess differences in potential biological functions between the two ferroptosis clusters.

### 4.8. External Validation of the Ferroptosis Clustering 

To further illustrate the relationship between ferroptosis clustering and clinical practice, we performed external validation. A recent study collected clinical information and synovial RNA expression profiles before and after anti-TNF therapy in RA patients [40]. Among the 46 rheumatoid arthritis patients studied, 19 had a good response, 13 had a moderate response and 14 had no response to the therapy. To assess the relationship between ferroptosis clustering and treatment response, we downloaded synovial RNA sequencing data from 19 good responders and 14 non-responders before anti-TNF treatment from the GEO database (accession number GSE 198520) and performed clustering analysis as previously described. Finally, we compared the proportions of different ferroptosis clusters between the good responders and non-responders to determine if there was an association between ferroptosis clustering and anti-TNF treatment response.

### 4.9. Construction of the Diagnostic Model

RA samples were randomly divided into two groups: the training cohort (70%) and the validation cohort (30%). The glmnet R package was used to construct the LASSO regression model in RA patients based on DEGs between ferroptosis clusters. Correlation coefficients were used to calculate the risk score as follows: Risk score = ∑i Coefficientsi × Expression level of signaturei. RA patients with higher risk scores were more likely to be classified into cluster 2, while those with low risk scores were the opposite. The diagnostic ability of this model was evaluated from the AUC value using the plotROC R package. GO and KEGG enrichment analyses were used to analyze the biological function of genes involved in the model.

### 4.10. Statistical Analysis

All statistical analyses were performed using R version 4.1.0 (R Foundation for Statistical Computing, Vienna, Austria). The Wilcoxon test was used for statistical analysis between the two groups. Pearson’s correlation method was used to predict the relationship between FRGs, and the correlation between FRGs and immune cells. A *p* value < 0.05 was considered statistically significant. 

## 5. Conclusions

In this study, we revealed and characterized the participation of FRGs in the pathogenesis of RA. Furthermore, we identified two distinct RA subtypes, exhibiting different states of immunity and ferroptosis. The results of our external validation suggested that RA patients in the two ferroptosis clusters responded differently to anti-TNF therapy. Finally, we established a diagnostic scoring system to classify individual RA patients. To the best of our knowledge, this is the first study to comprehensively analyze the relationship between ferroptosis and RA. Our results are consistent with previous mechanism studies. Although it is not easy to obtain synovium, large samples are still needed to verify RA classification in the future.

## Figures and Tables

**Figure 1 ijms-24-07342-f001:**
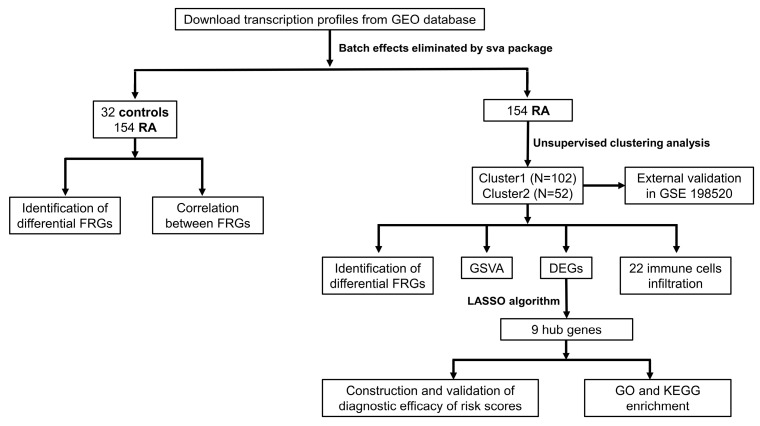
Workflow diagram of the study.

**Figure 2 ijms-24-07342-f002:**
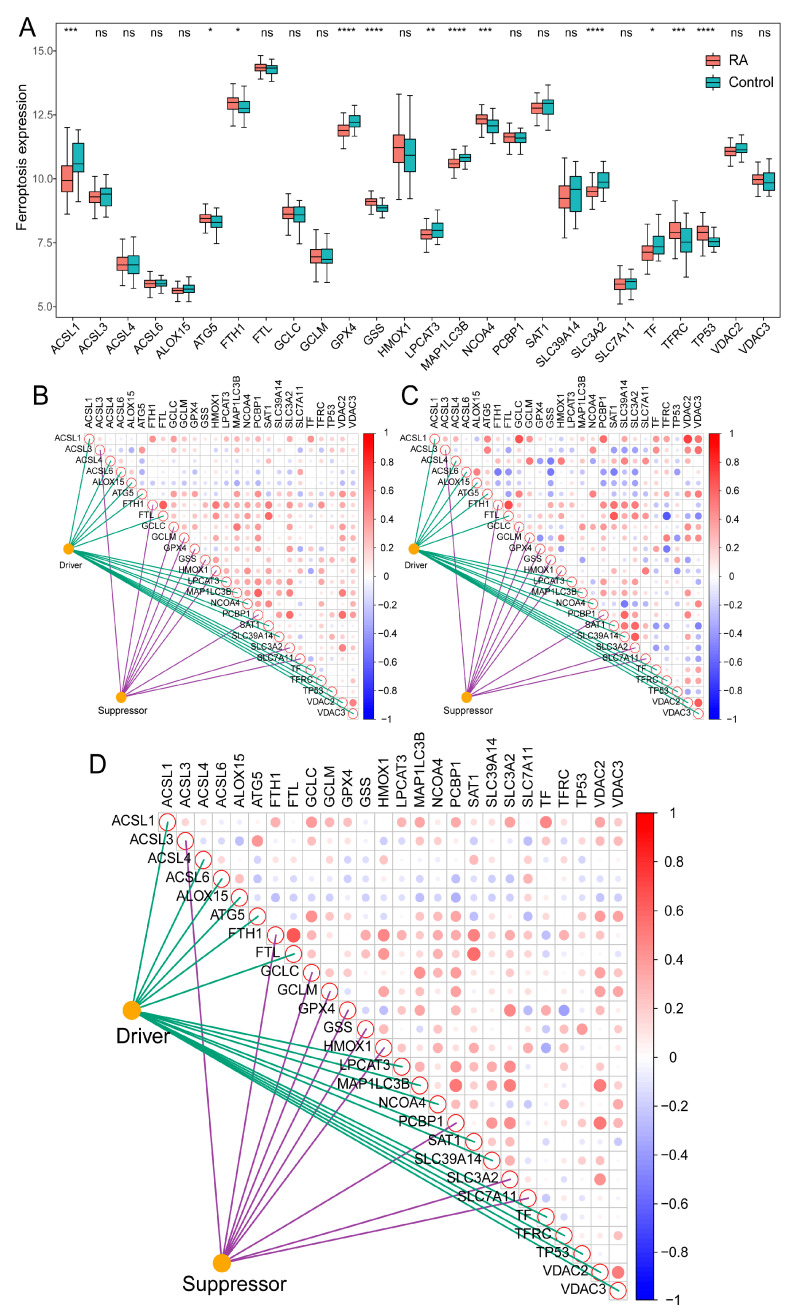
Landscape of FRGs in RA patients. (**A**) Degrees of expression of 26 FRGs in the RA and HC groups. (**B**–**D**) Correlation heatmaps of 26 FRGs in RA patients (**B**), HCs (**C**), and all samples (**D**). Positive correlation is shown in red, and negative correlation in blue; the size of the circle represents the intensity of correlation. *: *p* < 0.05; **: *p* < 0.01; ***: *p* < 0.001; ****: *p* < 0.0001; ns: not significant.

**Figure 3 ijms-24-07342-f003:**
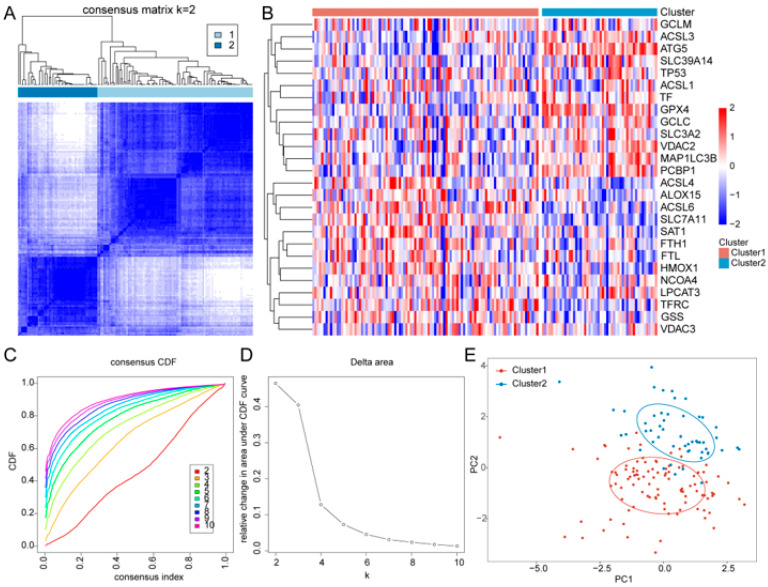
Identification of two ferroptosis clusters. (**A**) The consensus cluster matrix at k = 2. (**B**) Heatmap of 26 FRGs between the two ferroptosis clusters. (**C**,**D**) Relative change in the CDF delta area curve for k = 2–10. (**E**) Visualization of the clustering results based on PCA.

**Figure 4 ijms-24-07342-f004:**
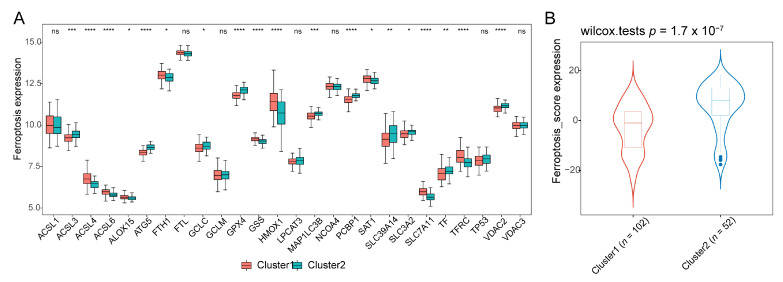
Differences in ferroptosis between the two RA patient clusters. Expression of 26 FRGs (**A**) and ferroptosis scores (**B**) in the two ferroptosis clusters. *: *p* < 0.05; **: *p* < 0.01; ***: *p* < 0.001; ****: *p* < 0.0001; ns: not significant.

**Figure 5 ijms-24-07342-f005:**
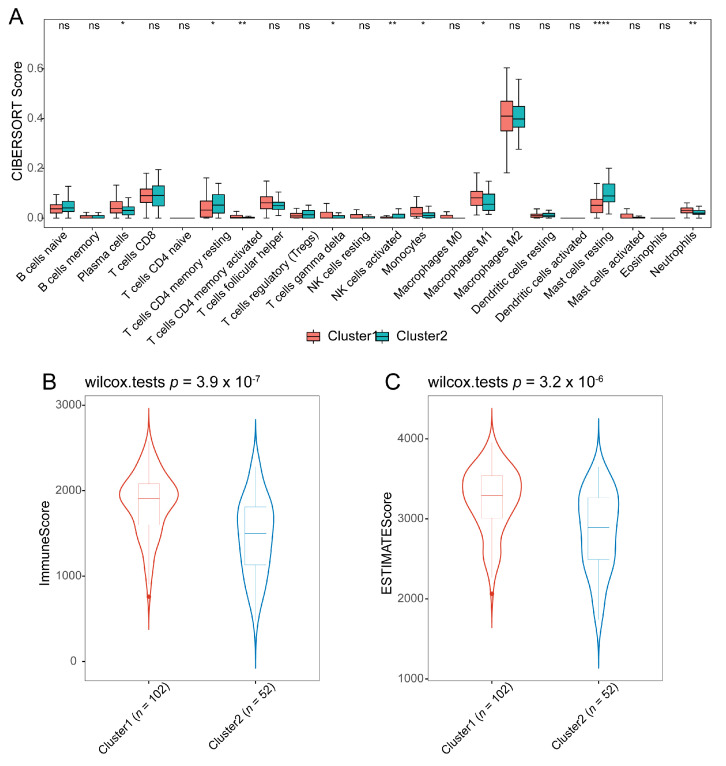
Differences in immune status in the two ferroptosis clusters. Proportions of 22 immune cell types (**A**) and estimated immune scores (**B**,**C**) in the two ferroptosis clusters. *: *p* < 0.05; **: *p* < 0.01; ****: *p* < 0.0001; ns: not significant.

**Figure 6 ijms-24-07342-f006:**
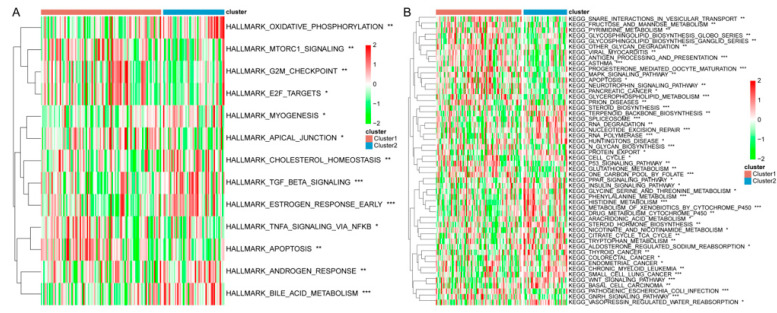
Biological enrichment analysis to further distinguish the two ferroptosis clusters. Hallmark Gene Set enrichment (**A**) and activated pathways (**B**) in cluster 1 (red) and cluster 2 (blue). *: *p* < 0.05; **: *p* < 0.01; ***: *p* < 0.001.

**Figure 7 ijms-24-07342-f007:**
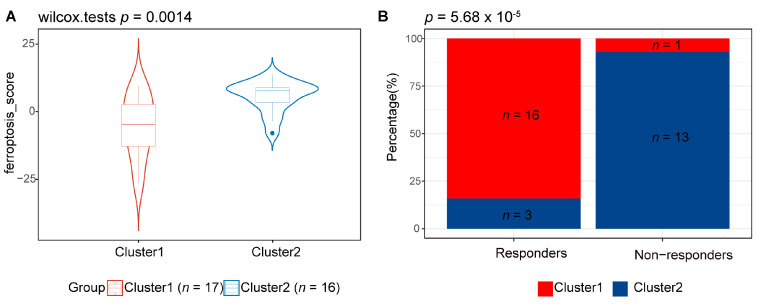
RA patients in cluster 1 responded better to anti-TNF therapy. Ferroptosis score in the two ferroptosis clusters (**A**) and amounts of cluster 1 and cluster 2 in responders and non-responders (**B**).

**Figure 8 ijms-24-07342-f008:**
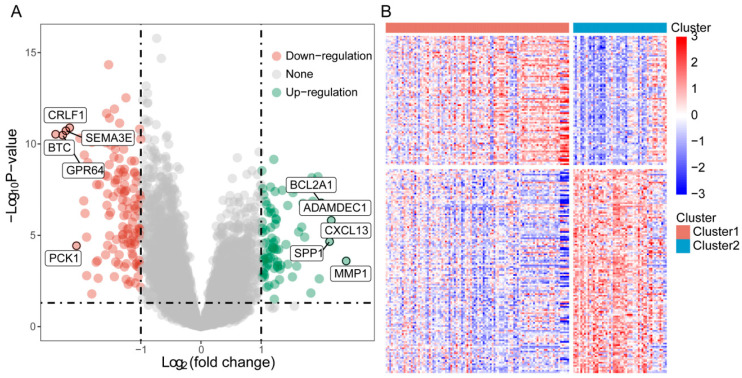
DEGs between ferroptosis clusters 1 and 2. DEGs are displayed as a volcano map, with blue representing upregulated genes and red representing downregulated genes in cluster 1 (**A**) and a heatmap (**B**).

**Figure 9 ijms-24-07342-f009:**
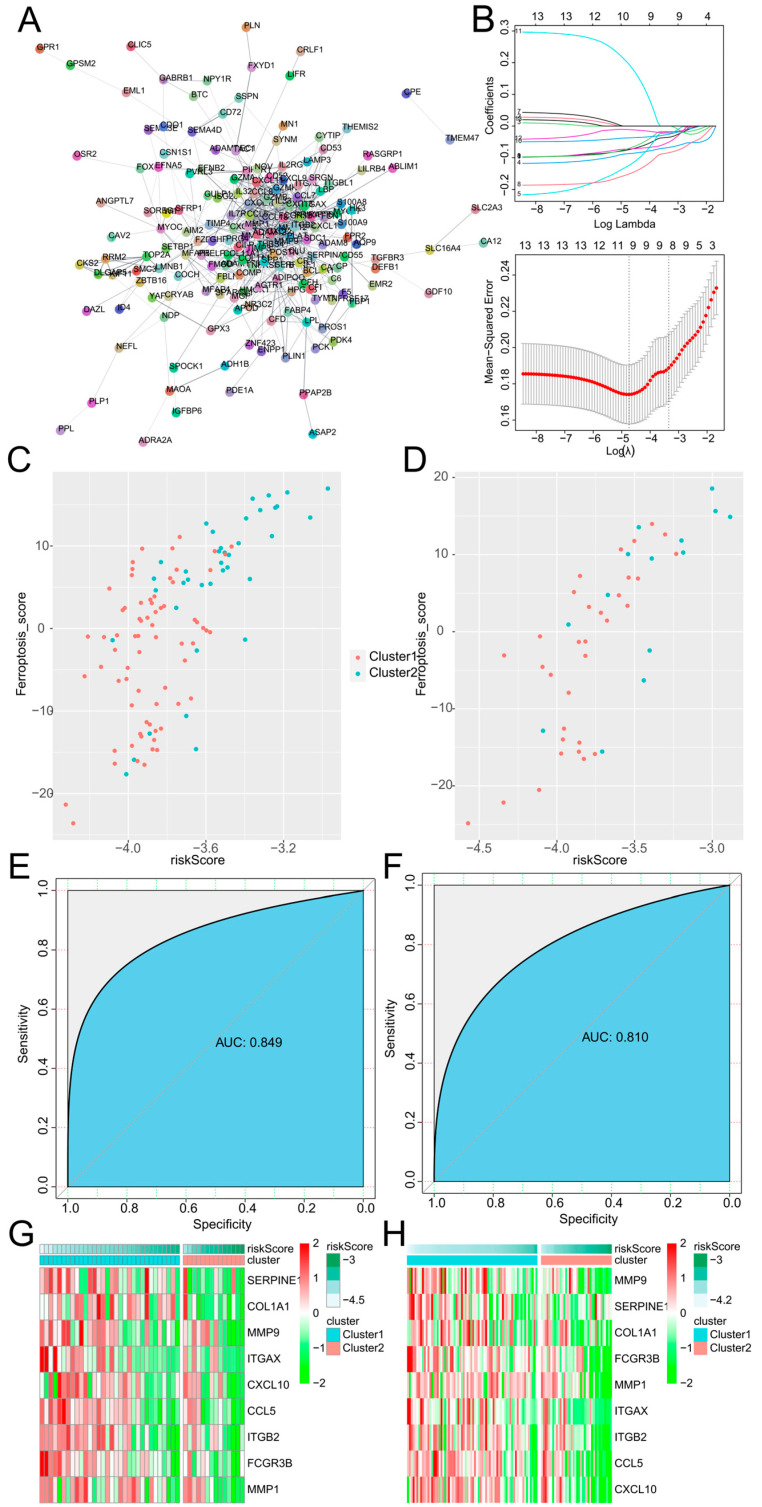
Diagnostic model for distinct ferroptosis clusters in RA. (**A**) PPI network of DEGs in ferroptosis clusters 1 and 2. (**B**) Construction of a diagnostic model based on hub genes. (**C**,**D**) Risk scores in the training and validation cohorts. (**E**,**F**) AUC values in the training and validation cohorts. (**G**,**H**) Heatmaps of hub gene expression in the training and validation cohorts.

**Figure 10 ijms-24-07342-f010:**
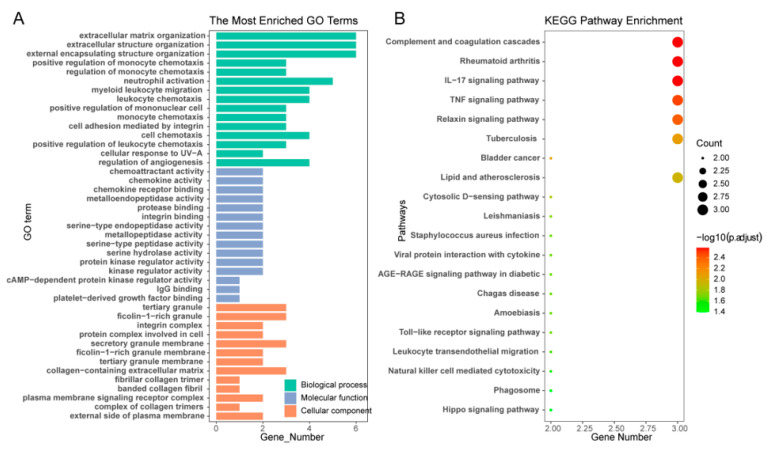
Enrichment analysis of nine diagnostic hub genes in the GO (**A**) and KEGG (**B**) databases.

**Table 1 ijms-24-07342-t001:** Data sources and platforms used to analyze synovial tissues from RA patients and HCs.

GSE_ID	Samples	RA	Control	Platform
GSE 153015	10	10	0	GPL570
GSE 1919	10	5	5	GPL91
GSE 36700	7	7	0	GPL570
GSE 55235	20	10	10	GPL96
GSE 55457	23	13	10	GPL96
GSE 55584	10	10	0	GPL96
GSE 77298	23	16	7	GPL570
GSE 48780	83	83	0	GPL570
Total	186	154	32	

## Data Availability

The datasets used in this investigation are available at the GEO database (https://www.ncbi.nlm.nih.gov/geo/, accessed on 2 October 2022). The names of the repository/repositories and accession number(s) can be found in the article/Appendix A.

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
