# Peer review of "Ferroptosis-Related Molecular Clusters and Diagnostic Model in Rheumatoid Arthritis"

_ijms, 2023, doi:10.3390/ijms24087342_

Round 1
Reviewer 1 Report
Dear Authors,
Please find comments in the attached file.
Kind Regards

Reviewer 2 Report
ijms-2264761
Ferroptosis-Related Molecular Clusters and Diagnostic Model in Rheumatoid Arthritis
This manuscript focuses on the impact of ferroptosis-related genes (FRGs) on the pathogenesis of rheumatoid arthritis (RA) and the definition of RA subtypes. Based on extensive bioinformatic analyses, the authors report that 12 (out of 26) FRGs were differentially expressed among RA patients and healthy controls (HC). According to the FRG expression profiles, RA patients could be divided into two clusters characterized by different ferroptosis scores and presumably showing different ferroptosis patterns. Moreover, for both clusters, the involvement of specific immune cells was predicted and signaling pathways with differentially expressed signaling molecules were identified. Finally, following the identification of differentially expressed genes (DEGs) between both ferroptosis clusters, the generation of a protein-protein interaction network, and the identification and selection of hub genes, a diagnostic model for the calculation of a risk score was generated. The authors conclude that their diagnostic scoring system may facilitate a more precise treatment for RA patients.
The article is well written and covers an interesting topic. The authors have clearly made an effort to address the impact of ferroptosis-associated gene clusters on rheumatoid arthritis via a variety of bioinformatics approaches. However, there are critical points/questions requiring the authors’ consideration.
Major comments:
In my opinion, the study has two major problems.
1. The bioinformatic analyses are not connected to medical/biological data. For instance, it is not clear whether (i) RA patients and HC or (ii) the RA patients who were divided into two clusters according to differentially expressed FRGs really show differing ferroptosis phenotypes (or any other clinically relevant differences). It also remains unclear whether the ferroptosis score reflects detectable (clinical) differences among the respective RA patients, whether patients in different clusters are indeed characterized by a differential involvement of the predicted immune cells, and whether the identified signaling pathways are differentially activated. Therefore, it remains unclear whether the bioinformatic results indicate manifest clinical differences among the respective patients.
2. To me, the connection between the initial analysis of the FRGs and the finally presented diagnostic model remains opaque. It is comprehensible that ‘usual suspects’ like MMP-9, CCL5, COL1A1, CXCL10, and Serpin E1 (and so on) are involved in RA pathogenesis and may thus contribute to a diagnostic model for RA (subtypes). However, does the development of such a model really presuppose the definition of ferroptosis-related clusters? In which way is the process of ferroptosis associated with the model and the genes contributing to the model? What does lower or higher risk mean in that case (since all patients are already suffering from RA)? Are there clinical differences among the defined subtypes? Which advantage for RA diagnosis or treatment does the definition of such subtypes provide? In the Discussion, the authors offer some speculation on some of these issues (e.g., concerning the response to anti-TNF therapy), but there is no robust evidence provided to support these assumptions.
Minor comments:
3. All abbreviations have to be defined in the text.
4. Though the composition of FRGs is referenced in 4.2 and shown in Figure 2, it would be helpful for the reader to better introduce origin, function, and selection of the FRGs used in this study.
5. Please better define the meaning of the ferroptosis score.
6. 2.4/Figure 5A: Since the differences between both clusters indicated in Figure 5A are rather small in most cases (though in part significant), the statement that “… cluster 1 had greater abundance of plasma cells, … and neutrophils. Meanwhile, cluster 2 had higher infiltration of resting memory CD4+ T cells, … and resting mast cells …” appears exaggerated.
7. 2.5: Please better explain the connection of the identified signaling pathways to ferroptosis. In which way do the identified pathways contribute to characteristics of the clusters? Please also provide a list of affected genes within these pathways (e.g., in the Supplement). Do the affected genes have activating or deactivating roles within the pathways?
8. Discussion (lines 186-187): The authors state that they “… have constructed a convincing diagnostic model to evaluate individual patients”. However, in respect of the data presented in Figure 8 (panels C and D), the risk scores of the individual patients within both clusters are quite mixed up, i.e., there is no sharp separation. Please comment on that.
9. In Table 1, a total of 186 samples is indicated though 196 samples are listed. Please correct. Moreover, for GSE153015, 20 samples are indicated including 10 RA and 0 HC samples. Where do the remaining 10 samples come from?
10. In general, it is hard to read the text in the figures. Please improve.
11. The References have to be adjusted to the IJMS style.
Round 2
Reviewer 2 Report
ijms-2264761
This manuscript provides a revised version of the study “Ferroptosis-Related Molecular Clusters and Diagnostic Model in Rheumatoid Arthritis”. The manuscript has been improved considerably and my comments have been adequately addressed.
Author Response
Dear Reviewer, Thanks for your comments. With the help of your insightful suggestions, the manuscript has been considerably improved. kind regards Maosheng Xie